# Optimization and Fabrication of an MOEMS Gyroscope Based on a WGM Resonator

**DOI:** 10.3390/s20247264

**Published:** 2020-12-18

**Authors:** Dunzhu Xia, Bing Zhang, Hao Wu, Tao Wu

**Affiliations:** 1School of Instrument Science and Engineering, Southeast University, Nanjing 210096, China; 220193324@seu.edu.cn; 2Huaihai Industries Group, Changzhi 046000, China; df11g25t@126.com; 3School of Information Science and Technology, ShanghaiTech University, Shanghai 201210, China; wutao@shanghaitech.edu.cn

**Keywords:** MOEMS gyroscope, WGM resonator, quality factor improvement, MEMS fabrication process

## Abstract

In this paper, the characterization of a whispering gallery mode (WGM) resonator applied in a novel micro-opto-electro-mechanical system (MOEMS) gyroscope was investigated. The WGM optical transmission coupling model was analyzed and compared by adjusting key parameters, such as the cavity radius, the waveguide width, and the gap between them for silicon and silicon nitride materials in simulations, which will greatly affect the quality factor (Q) of the WGM resonator. Furthermore, the structural parameters of the disk resonant gyroscope were also optimized. Then, the fabrication process was optimized to overcome the difficulties in the realization of micro-optical devices. Finally, a gyroscope prototype with the integrated WGM resonator was verified experimentally. The scale factor and bias instability performance of the MOEMS gyroscope was 2.63 mv/°/s and 4.0339°/h, respectively.

## 1. Introduction

Gyroscopes are essential devices for inertial measurement, which have been widely used for navigation, aviation, aerospace, and electronic warfare. In the last few decades, a rapid development in micro-electro-mechanical system (MEMS) technology has made the MEMS gyroscopes the dominating species in the consumer market. Although there are many benefits of MEMS gyroscopes, such as miniaturization, low cost, and ease of fabrication, their drawbacks cannot be negligible. Conventional MEMS gyroscopes adopt capacitance detection for angular velocity or angle changes (due to a change in the voltage of the sense electrodes). They are vulnerable to parasitic interference and have suboptimal accuracy. Fortunately, optical detective gyroscopes do not suffer from the same disadvantages and can be much superior in terms of sensitivity and anti-electromagnetic interference.

Conventional micro-opto-electro-mechanical system (MOEMS) gyroscopes are based on the Sagnac effect, which include ring laser gyroscopes and fiber optic gyroscopes [1,2,3]. Although the Sagnac gyroscopes are commercially mature, the centimeter-order size is a challenging task to combine with MEMS or an integrated circuit (IC). In 1910, Lord Raleigh reported in his work that the transmission distance of sound in St. Paul’s Church was particularly long; therefore, it was named as a whispering gallery mode (WGM) [4]. A typical WGM resonator consists of a straight waveguide and a resonant cavity. The gap between the straight waveguide and resonant cavity is small, which can be coupled through an evanescent wave. Common cavity shapes, such as a microsphere, microdisk, microring, and microcylinder, have various merits and demerits. Among these different shapes, microdisk and microring are more compatible with MEMS fabrication technology and have been actively pursued for chip-scale silicon photonics [5,6,7,8]. These resonators are ideal candidates for a variety of applications, including biosensors, vibration gyroscopes, and accelerometers, due to their miniaturization and high Q value. In 2008, researchers from University of Trento reported sub-nanometer microdisk resonators, which have quality factors of 2800 around the wavelength of 800 nm [9]. Moreover, bias instability of 0.41°/s over a one-hour test and quality factor of 3.85 × 10^6^ were realized with the help of a silica waveguide ring resonator [10]. Later on, a kind of cavity with a diameter of 3 mm, bias instability of 0.22°/s, and quality factor of 6.13 × 10^6^ was also reported [11]. The microdisk can potentially realize the high transmission efficiency of energy and high Q value due to its travelling-wave resonator. Furthermore, the fabrication cost of a micron/submicron-sized microdisk is much lower than a microring using electron beam lithography (EBL). The Kerr noise in the WGM-based gyroscope has also been analyzed, which can be well suppressed in the case of mode splitting [12]. In our previous work, we proposed an MOEMS gyroscope based on a WGM resonator and simulated its operational principle [13].

In this work, the WGM resonator was further optimized and verified. The disk resonator gyroscope (DRG) structure was chosen as its mechanical part, and the microdisk optical resonator as the sensitive detection part. The effect of different loss mechanisms and the key parameters of the WGM resonator were analyzed and optimized with FDTD Solutions software. After, we explored the processing technology of the sub-micron structure, and then fabricated the MOEMS gyroscope integrated with a WGM resonator. Finally, a series of performance tests were conducted to verify that this MOEMS gyroscope based on the WGM resonator can achieve a certain accuracy, which can be comparable with other kinds of MOEMS gyroscopes.

## 2. Coupling Mechanism of the WGM Resonator

### 2.1. Resonance Analysis of the WGM Resonator

Figure 1 shows a two-dimensional model of a WGM resonant cavity. When the incident angle of light satisfies the condition of total internal reflection, the light is completely confined to resonate in the cavity, and its propagation trajectory can be approximated as a polygon. The expression of the resonance light wavelength can be described as:(1)$$\lambda =\frac{2\pi Rn_{eff}}{l},$$
where λ is the matched wavelength, R is the radius of the cavity, $n_{eff}$ is the effective refractive index, and l is an integer, which represents the order of resonance.

As an electromagnetic wave, light can be described by Maxwell equations. Maxwell equations in the complex form are performed by the curl operation, which can be solved:(2)$$\nabla^{2}E=\omega^{2}\mu \varepsilon E,$$
where *E* is the electric field intensity, *ε* is the permittivity, and *μ* is the permeability.

To further describe the electromagnetic field distribution in the WGM resonator, a three-dimensional cylindrical coordinate system in Figure 2 is established, and the three vectors are represented by *r*, *φ*, and z.

In the ideal case of passive and no dielectric loss, the general form of the Helmholtz equation can be written as:(3)$$E(r,\varphi ,z)=a_{r}E_{r}(r,\varphi ,z)+a_{\varphi }E_{\varphi }(r,\varphi ,z)+a_{z}E_{z}(r,\varphi ,z).$$

The wave equation can be expanded into three scalar partial differential equations:(4)$$\begin{array}{l} \nabla^{2}E_{r}-(\frac{E_{r}}{r^{2}}+\frac{2}{r^{2}}\frac{\partial E_{\varphi }}{\partial \varphi })=-\omega^{2}\mu \varepsilon E_{r}\\ \nabla^{2}E_{\varphi }-(\frac{E_{\varphi }}{r^{2}}+\frac{2}{r^{2}}\frac{\partial E_{r}}{\partial \varphi })=-\omega^{2}\mu \varepsilon E_{\varphi }\\ \nabla^{2}E_{z}=-\omega^{2}\mu \varepsilon E_{z} \end{array}$$

The z-component partial differential equation can be solved:(5)$$\begin{array}{l} \nabla^{2}F=-\omega^{2}\mu \varepsilon F=\frac{\partial^{2}F}{\partial r^{2}}+\frac{1}{r}\frac{\partial F}{\partial r}+\frac{1}{r^{2}}\frac{\partial^{2}F}{\partial \varphi^{2}}+\frac{\partial^{2}F}{\partial z^{2}}\\ F=\{E\} \end{array}$$

For the microdisk resonator, the polarization directions of the TE mode and the TM mode are parallel and vertical to the surface of the microdisk resonator. By substituting the separation variable $F_{z}=\Psi (r)\Phi (\varphi )Ζ(z)$ into Equation (5), we can derive:(6)$$\Phi Ζ\frac{\partial^{2}\Psi }{\partial r^{2}}+\Phi Ζ\frac{1}{r}\frac{\partial \Psi }{\partial r}+\Psi Ζ\frac{1}{r^{2}}\frac{\partial^{2}\Phi }{\partial \varphi^{2}}+\Psi \Phi \frac{\partial^{2}Ζ}{\partial z^{2}}=-\omega^{2}\mu \varepsilon \Psi \Phi Ζ.$$

Let:(7)$$\begin{array}{l} \frac{\partial^{2}Ζ}{\partial z^{2}}=-\beta_{z}^{2}Ζ\\ \frac{\partial^{2}\Phi }{\partial \varphi^{2}}=-l^{2}\Phi \end{array}$$

Thus, the typical Bessel function can be described as:(8)$$r^{2}\frac{\partial^{2}\Psi }{\partial r^{2}}+r\frac{\partial \Psi }{\partial r}+[\beta_{r}^{2}r^{2}-l^{2}]\Psi =0,$$
where $\beta =\omega^{2}\mu \varepsilon$, and $\beta_{z}^{2}+\beta_{r}^{2}=\beta^{2}$. The general solution of the above equation can be expressed as:(9)$$\Psi =AJ_{l}(\beta_{r}r)+BY_{l}(\beta_{r}r),$$
where $J_{l}$ and $Y_{l}$ are the first kind and the second kind of Bessel function, respectively. A and B depend on the boundary conditions.

Since the size of the microdisk resonator in this paper is very small, the latter term in Equation (9) can be ignored. The evanescent field decays exponentially outside the microdisk resonator:(10)$$\psi =\left\{\begin{array}{ll} AJ_{l}(\beta_{r}r)r\leq R & r\leq R\\ J_{l}(\beta_{r}r)\cdot e^{\left(-\frac{2\pi \sqrt{n_{eff}{}^{2}-n_{0}^{2}}\left(r-R\right)}{\lambda }\right)} & r>R \end{array}\right.,$$
where R is the radius of the microdisk resonator.

### 2.2. WGM Resonator-Waveguide Coupling System

The model of cavity-waveguide coupling and its detecting principle is shown in Figure 3. During the transmission of light E1 along the waveguide, a certain frequency is limited in the cavity due to the evanescent wave phenomenon [14].

The coupled equation of the model can be expressed as:(11)$$\left(\begin{array}{c} E_{2}\\ E_{3} \end{array}\right)=\left(\begin{array}{cc} t & -\kappa^{*}\\ \kappa & t^{*} \end{array}\right)\left(\begin{array}{c} E_{1}\\ E_{4} \end{array}\right),$$
where t is the self-coupling coefficient, and κ is the mutual coupling coefficient of the waveguide and cavity. Then:(12)$$t=\left|t\right|e^{j\phi },$$
where ϕ is the phase mismatch caused by coupling. In the case of no loss, the coupling coefficient satisfies:(13)$${\left|t\right|}^{2}+{\left|\kappa \right|}^{2}=1.$$

Due to phase accumulation and amplitude attenuation, the coupled field in the resonator meets:(14)$$E_{4}=\alpha e^{j\theta }E_{3},$$
where α and θ are the losses and phase shifts caused by propagation.

The normalized transmissivity can be written as:(15)$$T={\left|\frac{E_{2}}{E_{1}}\right|}^{2}=\frac{\alpha^{2}+{\left|t\right|}^{2}-2\alpha \left|t\right|\cos(\phi +\theta )}{1+\alpha^{2}{\left|t\right|}^{2}-2\alpha \left|t\right|\cos(\phi +\theta )}.$$

In theory, since the coupling gap is short enough, the influence of the phase mismatch ϕ can be ignored in the first order.

When resonance occurs in the resonator (ϕ+θ=2nπ, n is a positive integer), there are three kinds of coupling states:

(1) Under-coupling state: when the waveguide is far away from the WGM resonator, the coupling is weak. At this time, $\left|t\right|>\alpha$, the transmissivity cannot reach zero.

(2) Critical coupling state: As the waveguide gets closer to the WGM resonator, resonance occurs and when $\left|t\right|=\alpha$ is satisfied. The transmissivity is equal to zero.

(3) Over-coupling state: when the distance between them is closer, the transmissivity is greater than zero again.

### 2.3. Quality Factor of the WGM Resonator

The quality factor (*Q*) is an important indicator to measure the performance of the WGM resonator. The *Q* is related to the photon lifetime τ and can be roughly estimated by the transmission line. It can be expressed by the following equation:(16)$$Q=\omega_{r}\tau .$$

The reciprocal of the *Q* represents the loss of the resonator. $\omega_{r}$ is the resonant angle frequency. The total loss $Q^{-1}$ is caused by a variety of factors, such as [15]:(17)$$Q^{-1}=Q_{cp}^{-1}+Q_{rad}^{-1}+Q_{sca}^{-1}+Q_{bulk}^{-1},$$
where $Q_{cp}^{-1}$ is due to the coupling of the straight waveguide and cavity, $Q_{rad}^{-1}$ is related to the radius loss, $Q_{sca}^{-1}$ depends on the waveguide roughness, and $Q_{bulk}^{-1}$ depends on the material property.

## 3. Analysis and Optimization of the MOEMS Gyroscope

The optimization of the MOEMS gyroscope is focused on in this section, which mainly includes the optimization of the WGM resonator and the DRG structure. The angular velocity detection is changed from a conventional capacitive method to an optical one, which is finally converted into the light intensity output.

### 3.1. Optimization of WGM Resonator

The finite-different time-domain (FDTD) method is a numerical analysis method that can model and calculate electromagnetic fields. The FDTD method is based on solving the Maxwell equations related to time. By discretizing the wave equations in the time domain, the Maxwell curl equations are converted into difference equations. The distribution of the electromagnetic field can be derived through repeated iterations by FDTD Solutions.

The cladding layer is selected as silica with a lower refractive index (*n* = 1.44) and a thickness of 2 μm. Two different refractive index materials, silicon (*n* = 3.48) and silicon nitride (*n* = 2), are compared and analyzed. Silicon and silica have a large refractive index difference, thus most of the energy can be restricted into the waveguide, which means that the absorption loss is extremely small. Besides, the fabrication method for silicon is mature enough to etch high aspect ratio structures on it. However, silicon is sensitive to scattering loss, which is inevitable during MEMS fabrication. In contrast, although silicon nitride is slightly inferior to silicon in terms of the ability to confine light, the smaller refractive index difference ensures that the roughness has less of an effect on the device.

According to Equation (17), $Q_{rad}^{-1}$ and $Q_{cp}^{-1}$ can be optimized by adjusting the structure parameters. Since the high-order modes in the microdisk resonator are more obvious, the radius of the resonator is generally less than 10 μm. When the thickness of the silicon waveguide is 0.2 μm and the width changes from 0.4 to 0.5 μm, the optical field can be transmitted through the waveguide in a single mode. For silicon nitride with a small refractive index, the thickness and width of the waveguide should be appropriately increased. The model parameters of the simulation are set as in Table 1. The radius of the resonator is continuously changed by a step of 1 μm. The relationship between the transmission line and the resonator radius is shown in Figure 4.

It can be seen in Figure 5 that the coupling state is closer to the critical coupling state as the radius increases; meanwhile, the higher-order modes in the cavity become obvious, and even occupy the main mode. When the radius of the silicon resonator is 4 μm, there is only one mode, while the silicon nitride resonator always has more than 2 modes. According to the results, the radius of the silicon and silicon nitride resonator is selected as 4 μ and 5 μm, respectively.

The important parameter that affects $Q_{cp}^{-1}$ is the coupling gap. The transmission line and quality factor changes under different gaps are shown in Figure 6 and Figure 7. As the coupling gap increases, the coupling between the resonator and the waveguide becomes weaker, and the system is under-coupled. Meanwhile, the quality factor increases as the system coupling weakens. Since the quality factor calculation of the high-Q cavity in FDTD Solutions is inversely proportional to the slope of the attenuation envelope of the energy field, the smaller energy decay causes a higher Q value. The quality factor of silicon is higher than that of silicon nitride, but the consistency of the transmittance of silicon nitride is better than that of silicon under different gaps. Considering the coupling efficiency and Q value, the coupling gap of the silicon and silicon nitride are 0.2 and 0.15 μm, respectively.

Besides, the waveguide width can also influence the coupling effect in Figure 8 and Figure 9. It can be seen that from a silicon resonator that the critical coupling occurs when the waveguide width is 0.4 μm, while for a silicon nitride resonator, the value is 0.5 μm. When the width of the silicon waveguide is less than 0.4 μm, the resonance wavelength shifts. Therefore, the quality factor cannot be calculated as the waveguide width is 0.35 μm. However, when the width of the silicon waveguide is greater than 0.4 μm, the light at the resonance point cannot be completely confined in the resonator. The consistency of transmittance of silicon nitride is also better than that of silicon.

In conclusion, the final parameters are selected as shown in Table 2.

$Q_{sca}^{-1}$ depends on the roughness of the waveguide. Because the conventional dry etching processes are based on the “etch-passivation” principle, the roughness during etching cannot be ignored. According to the research results by Lacey and Payne, the scattering loss α can be expressed as [16]:(18)$$\alpha =4.34\frac{\sigma^{2}}{\sqrt{2}\kappa_{s}W^{4}n_{clad}}g(V)\xi (\chi ,\gamma ),$$
where σ is roughness root mean square (RMS) of the waveguide, $\kappa_{s}$ is the reciprocal coefficient of the wavelength in centimeters, W is the diameter of the resonator, $n_{clad}$ is the reactive index of the cladding layer, and g(V) and ξ(χ,γ) are slowly changing functions that have little effect on loss.

The size of the silicon nitride resonator is larger, and its resonance wavelength is smaller. Therefore, the scattering loss should be smaller. To further determine the effect of roughness on different materials, the sidewall roughness is introduced to the structure. The roughness (RMS) changes from 5 to 20 nm, and the simulation results are shown in Figure 10.

As illustrated in the Figure 10, since the transmissivity varies greatly with different roughness, the tiny roughness of the sidewall will have an influence on the performance of the silicon device. In contrast, the silicon nitride material is much less affected. According to the simulation results, the performance of the silicon devices under ideal conditions is much better than that of silicon nitride. However, the error in the actual fabrication is inevitable. The stability of silicon nitride is better than silicon; therefore, silicon nitride was selected as the fabrication material in this work.

### 3.2. Analysis of the Mechanical Part

The structure of the DRG is shown in Figure 11. We adopted the disk resonant gyroscope structure proposed by Stanford University for reference, which was also optimized [17]. The mechanical part of the MOEMS gyroscope is composed of an anchor, spokes, and multiple sets of concentric rings.

According to the Coriolis effect, the dynamic equations of the DRG in the wine glass mode can be expressed as typical second-order differential equations [18]:(19)$$\begin{array}{l} \ddot{x}+2\xi_{x}\omega_{x}\dot{x}+\omega_{x}^{2}x=\frac{F_{0}\sin(\omega_{x}t)}{m_{eff}}\\ \ddot{y}+2\xi_{y}\omega_{y}\dot{y}+\omega_{y}^{2}y=4A_{g}\Omega \dot{x} \end{array}$$
where x and y represent the displacement in the driving direction and sense direction, $\omega_{x}$ and $\omega_{y}$ are the resonance frequency in the drive and sense modes, $F_{0}\sin(\omega_{x}t)$ represents the applied driving force with the initial phase of 0, $m_{eff}$ is the equivalent mass, $A_{g}$ is the angular gain, and Ω is the angular velocity.

The solution of the displacement of the drive mode is:(20)$$x(t)=\frac{Q_{x}F_{0}}{k_{x}}\sin(\omega_{x}t-\frac{\pi }{2})=x_{0}\sin(\omega_{x}t-\frac{\pi }{2}),$$
where $Q_{x}$ is the quality factor of drive mode, and $k_{x}$ is stiffness on the driving axis. In theory, the drive and sense modes are perfectly matched ($\omega_{x}$ = $\omega_{y}$ = $\omega_{0}$), which the mechanical sensitivity can be expressed as:(21)$$\left|\frac{y_{0}}{\Omega }\right|=\frac{4A_{g}Q_{x}F_{0}}{\omega_{x}m_{eff}\sqrt{{\left(\omega_{y}^{2}-\omega_{x}^{2}\right)}^{2}+{\left(\frac{\omega_{x}\omega_{y}}{Q_{y}}\right)}^{2}}}=\frac{4A_{g}Q_{s}x_{0}}{\omega_{0}},$$
where $y_{0}$ is the displacement in the sense direction, and $Q_{y}$ is the quality factor of sense mode ($Q_{x}=Q_{y}{=Q}_{s}$).

The above equation describes the mechanical sensitivity, which can be improved by increasing the driving force, improving the quality factor, and reducing the frequency split. The structural optimization mainly improves the latter two parts. Previous existing research results show that, by fine-tuning the size parameters of the spokes and the ring, the effective stiffness can be adjusted to compensate for the anisotropy of <100> silicon, which in turn leads to a greater improvement of the quality factor and frequency split. The process of optimizing structural parameters is mainly divided into two parts. Firstly, we adjust the main parameters (spoke number, ring number, spoke length, and ring length) to meet the set quality factor, and then we adjust the offset angle to make the frequency split meet the set value [18]. Through the joint simulation of ANSYS and COMSOL, the main energy loss mechanism that affects the $Q_{s}$ value and the frequency split of the device is analyzed. The structure parameters are listed in Table 3. The resonance frequencies in the drive and the sense modes are 8324.8 and 8326.2 Hz, respectively.

As shown in the Figure 12, the WGM resonator is integrated on the outermost ring in the direction of the sensitive axis. When the disk of the gyroscope deforms, the WGM resonant cavity is driven to deform, which causes the resonant wavelength to shift. According to the second-order mode of the DRG, the shape of the cavity is approximated as an ellipse. The long and short semi-axes are assumed to be a and b, respectively. The shift of spectral line Δλ is:(22)$$\Delta \lambda =\frac{\Delta Ln_{eff}}{l}=\frac{[2\pi b+4(a-b)-2\pi R]n_{eff}}{l},$$
where ΔL is the perimeter change of the resonant cavity.

The sensitivity of the MOEMS gyroscope can be expressed as the ratio of spectral line drift to the input angular velocity. The coupling relationship can be divided into two parts, namely from angular velocity Ω to strain ε and from strain to spectral line drift Δλ. Similar to Equation (21), the relationship between the strain and input angular velocity can be expressed as:(23)$$\frac{\varepsilon }{\Omega }=\left|\frac{\Delta L}{L\Omega }\right|\approx \left|\frac{2\pi b+4(a-b)-2\pi R}{2\pi R\Omega }\right|=\frac{K_{xy}K_{y\lambda }}{r_{0}}\frac{4A_{g}Q_{s}x_{0}}{\omega_{0}}.$$

By combining Equations (22) and (23), the MOEMS sensitivity can be described as:(24)$$S=\left|\frac{\Delta \lambda }{\Omega }\right|=\left|\frac{\Delta L\lambda }{L\Omega }\right|\approx \left|\frac{2\pi b+4(a-b)-2\pi R}{2\pi R\Omega }\lambda \right|=\frac{K_{xy}K_{y\lambda }}{r_{0}}\frac{4A_{g}Q_{s}x_{0}}{\omega_{0}}\lambda ,$$
where $K_{xy}$ and $K_{y\lambda }$ are empirical coefficients related to the radius of the microdisk resonator, and for a WGM resonator with a radius of 5 μm, $K_{xy}$ corresponds to 2.301 × 10^−4^, and $K_{y\lambda }$ is equal to 0.4886.

A small angular velocity and a very short drift of the spectral line will require a high-resolution spectrometer. The detection range can be improved by converting the spectral line drift into light intensity changes. The input and output light intensity of the WGM resonator can be assumed to be I_1_ and I_2_, then the transmissivity T is:(25)$$T=\frac{I_{2}}{I_{1}}.$$

Since there is a largest spectral drift of the WGM resonant cavity at T = 1/4 in the transmission spectrum, when the resonant cavity is close to critical coupling, the following is satisfied [19]:(26)$$\frac{dT}{d\lambda }=1.28\frac{Q_{op}}{\lambda },$$
where $Q_{op}$ is the quality factor of the WGM resonant cavity, and the relationship between the input angular velocity and the change of output light intensity can be expressed as:(27)$$\frac{dI_{2}}{\Omega }=\frac{dT}{\Omega }I_{1}=\frac{1.28\cdot Q_{op}\cdot \frac{d\lambda }{\lambda }}{\Omega }I_{1}=\frac{1.28\cdot Q_{op}\cdot S}{\lambda }I_{1},$$
where *S* is the sensitivity of the MOEMS gyroscope. Therefore, the deviation of the spectral line can be converted into the change of the output light intensity through Equation (27). The transformation relationship is shown in Figure 13.

## 4. Fabrication Process

### 4.1. Design of the MEMS Fabrication Process

The key parameters of the WGM resonator were decided by the design and analysis. The MOEMS gyroscope is fabricated on an SOI wafer, where the top to bottom layers are the device layer of Si with a 40-μm thickness, a buried layer SiO_2_ with a 2-μm thickness, and a substrate layer Si with a 475-μm thickness. The Si layers are made of boron-doped <100> orientation single crystal silicon. The designed fabrication process is shown in Figure 14.

Figure 14a–i are the processing of the MOEMS gyroscope on the SOI substrate, which includes the processing of the gyroscope resonator and the micro-optical device, while Figure 14j–n are the glass processing and (o) is the result of bonding.

(a) The process started from an SOI wafer with a 2-μm-thick SiO_2,_ which was thermally grown on both sides. Then, a 300-nm-thick Si_3_N_4_ was deposited with the help of low-pressure chemical vapor deposition (LPCVD).

(b) The lithography was put on the bottom side to feature the hole area. The SiO_2_ and Si_3_N_4_ layers of the backside were etched by reactive ion etching (RIE), whereas the Si layer was etched by inductively coupled plasma (ICP) etch under the protection of photoresist, SiO_2_, and Si_3_N_4_. The wafer was immersed in an HF acid solution with the protection of the top layer to release the buried layer.

(c) Then, the optical structure of Si_3_N_4_ was left after the lithography and other materials were etched away using the RIE.

(d) After metal electrode deposition on the Si surface, the third lithography was performed to etch the SiO_2_ layer on the top side.

(e)–(g) The negative photoresist was spun on the surface for photolithography, and gold was plated on the surface by the magnetron sputtering process, in which chromium was used as the adhesion layer between the wafer and gold. Next, the lift-off processing was carried out to obtain the metal electrode and bonding area.

(h) The micro-optic device pattern was formed through electron beam lithography (EBL). The etching scheme was reasonably selected to etch the structure.

(i) Finally, the gyroscope structure was fabricated through ICP etching, like step (b).

(j) The glass was cleaned with a thickness of 500 μm, and then plasma-enhanced chemical vapor deposition (PECVD) under a lower working temperature was chosen to deposit Si_3_N_4_ on both sides.

(k) Patterns of electrode holes and grating holes on the front side were formed on the photoresist, while the glass cavities were masked by the gold on the backside.

(l) The glass was immersed in an HF solution to etch a shallow structure on both sides. After, a customized Teflon protection device was used to protect the front side, and then we continued to etch the backside using HF to obtain a deep glass cavity (approximately 20–30 μm).

(m)–(o) The shallow hole pattern was aligned with a microscope and a high-precision mobile platform, and then a diode pump laser was used to penetrate the electrode holes and leave about 20 μm for the grating holes. Finally, the bonding process of the SOI and glass were completed by gold-tin bonding.

A 3-D view of the designed MOEMS gyroscope is shown in Figure 15, where each layer is:
(A) Glass; (B) silicon nitride layer on SOI; (C) silica layer on SOI; (D) metal layer on SOI; and (E)–(F) device layer, buried layer, and substrate layer of SOI.

### 4.2. Optimization of the Fabrication Process on SOI

Step (a)–(i) are the processing on the SOI. There are two main difficulties, which are metal deposition and micro-optical device fabrication.

According to the fabrication process (g), in the case of metal electrode deposition on the Si surface, it is necessary to etch the silica on the SOI first. Since the RIE machine works for a longer period of time and can etch 2 μm SiO_2_, the increase in internal temperature causes possible denaturation of the photoresist on the wafer surface. In order to overcome this issue, we divided the RIE process into four times to complete. After, the lithography was performed to deposit 30-nm-thick Cr and 200-nm-thick Au on the Si surface, and then we used an additive kind of technique, namely lift-off. As shown in Figure 16a, the metal layer does not adhere well due to the large surface roughness. In order to ensure a smooth deposition area, the buffered oxide etch (BOE) was selected as an alternative scheme, while considering the etching rate and selection ratio of the mask. Figure 16b shows the results of better metal adhesion of the BOE-etched wafer.

For the processing of micro-optical devices, conventional lithography cannot meet the line width, thus electron beam lithography (EBL) was adopted [20]. EBL requires the electrical conductivity of the wafer. Since the cost of magnetron sputtering is high, and the corrosion of gold and chromium may affect the lithography pattern, it is not the first choice. An alternative solution is to use a water-soluble conductive adhesive to coat the electron beam photoresist. After the photolithography is completed, the conductive adhesive is washed away with water before development. Furthermore, the conductive adhesive can also eliminate the accumulation of charge during the exposure. The two conductive adhesives in this paper were AR-PC 5090 and AR-PC5091, which were used in combination with positive E-beam photoresist PMMA-A4 and negative E-beam photoresist AR-N 7520.

Scattering loss can greatly affect the device performance; therefore, it is necessary to reduce the roughness of the surface of a micro-optical structure. Many methods have been reported to solve this problem. One of these methods is the improved fabrication process, where the EBL is combined with ion beam etching (IBE), RIE, or some other wet methods. In this paper, three schemes were carried out to optimize the structure as shown in Figure 17. The micro-optic structure includes the grating coupler, waveguide, and microdisk.

#### 4.2.1. Scheme 1. Metal, PMMA-A4, IBE, and RIE

A very thin layer of Cr-Au was first deposited on the surface, as a conductive layer for later steps. After, the PMMA-A4 photoresist was coated on the entire surface and patterned by EBL. The scan field size was selected as 100 μm × 100 μm, the area dose as 150 μC/cm^2^, and the range of the dose factors from 0.4 to 1.1. Then, IBE, a kind of pure physical etching, was selected to etch the metal drain layer on the top. The results of different dose factors are listed in Table 4.

It can be concluded from the results that the amount of etching increases as the dose factor increases, and high does are required for the finer the etched lines. Since the radius of the microdisk is much larger than the etched lines, the $Q_{rad}$ value is almost unaffected. According to the results, the grating dose factor of 0.5 and the waveguide and cavity dose factor of 0.7 can be selected to obtain better etching results. After IBE, RIE was performed to etch the structure.

Figure 18a shows that due to the poor selectivity and uniformity of IBE, there are residual reactants in the etched trench. The trench after RIE is also inverted tapered, which may affect the performance (shown in Figure 18b).

#### 4.2.2. Scheme 2. PMMA-A4 and RIE

In order to avoid the difficulty of etching metal, we tried to perform RIE directly with the protection of E-beam photoresist. To enhance the corrosion resistance of PMMA-A4, the photoresist was post baked for 5 min. After etching, the roughness of the trench was significantly improved as shown in Figure 19. However, there are still two challenges, which are excessive lateral etching and a too-small etching depth.

The boundary between silica and silicon nitride is obvious in Figure 19, thus the etching depth should be above 300 nm. Due to the poor corrosion resistance of PMMA-A4, the silicon nitride layer was thinned by about half. Similarly, the excessive lateral etching is also due to the depletion of photoresist. Therefore, the reasonable choice of a protective layer is critical. Obviously, using PMMA photoresist as protection is not enough to meet the etching demand.

#### 4.2.3. Scheme 3. AR-N 7520, Metal, Lift-off, and RIE

Referring to the lift-off process in the metal deposition, AR-N 7520 negative photoresist was used for lithography. The total dose should be reduced about 10 times when selecting negative photoresist and taking the exposure parameters in Table 4 as a reference; therefore, the dose factor of the line was reduced to 0.43 and 0.63, respectively. The metal was evaporated by highly directional electron beam evaporation. To ensure the line accuracy, the thickness of the Cr-Au layer was set to 10 and 40 nm, respectively.

By continuously adjusting the parameters to balance lateral etching and smoothness, the etching result of RIE etching for 4 min and BOE etching for 15 min is shown in Figure 20. The roughness of the etched trench was significantly improved by adopting this scheme.

### 4.3. Optimization of the Fabrication Process on Glass

Since more wet etching processes are involved in the processing of glass, a pre-experiment was conducted to study the key parameters in wet etching and provide guidance for the compensation of the photolithography. After the electrode holes and grating holes on the front of the glass were tested, two sets of schemes were carried out. The AZ4620 photoresist was selected as the protective layer. The photoresist thickness of scheme I is 8 μm and the post-baking time is 2 min, while the photoresist thickness of scheme II is 12 μm and the post-baking time is 10 min. According to the results of the random test under two extreme cases, the data are summarized in Table 5 and Table 6.

The etching results under different parameters by HF and BOE solutions were obtained. The above data also verifies the feasibility of protecting the front of the glass with photoresist. The masking ability of the photoresist in scheme II is better. The horizontal and vertical etching ratio is also smaller. For shallow corrosion, the rate of BOE is relatively better controlled, so it is the preferred solution.

The glass cavity is a conformal octagon that is slightly larger than the DRG resonator, while the required etching depth is 20–30 μm. Obviously, BOE etching cannot meet its depth requirements. Therefore, HF was selected as the etching solution, and Cr-Au was used as the protective layer. Multiple sets of experiments were also conducted. The test data are illustrated in Table 7. Since Cr-Au will not fall off due to HF corrosion, the lateral etching of the glass under the protection of Cr-Au is significantly reduced.

In the design of the photolithography, the compensation is reasonably designed according to the pre-experimental data to ensure the integrity of the structure during wet etching. Referring to the experiment data, the cavity is etched in the HF solution by the protection of a gold mask, and the holes are etched in BOE solution only by the protection of the photoresist. Finally, the overall post-process of the glass structure is completed by laser drilling. Figure 21 shows the structure on SOI and glass. The SOI and glass are completed by gold-tin bonding. The light holes and the electrode holes are both aligned with the grating coupler and the electrodes.

## 5. Performance Test

### Experimental Setup

The MOEMS gyroscope was fabricated, packaged, and evaluated by experiments. The experimental setup is shown in Figure 22. The whole test system consists of two parts, including an optical detection loop and closed-loop drive circuit. The resonant capacitance value in drive mode is extracted by the C/V conversion circuit and amplified by an instrument amplifier. The phase and amplitude are controlled respectively by a precise 90° phase shifter and automatic gain control (AGC) module. The in-phase AC signal and the reversed signal are superimposed with a DC reference voltage $U_{d}$, respectively, which are applied on the electrodes to make the whole disk ring resonance.

The tunable laser emits a certain wavelength (λ = 1056 nm) of laser light into the polarization controller through the single-mode optical fiber, which ensures that the output light is in a single polarization state. The fiber and waveguide are extremely coupled by gratings. The photodetector is connected to the output end of the waveguide. When an angular velocity is exerted, the output light intensity will change correspondingly. Thus, the angular velocity can be deduced by detecting the output voltage of the photodetector, which is demodulated with the same frequency signal, i.e., the reference signal. The output data are finally collected by the multimeter and sent to the host computer for storage and processing. Meanwhile, we designed an electrical detection scheme as an experimental reference.

The open-loop test determines the resonance frequencies of the device for drive and sense modes. Firstly, the resonance frequency is searched over a range from 7.7 to 7.8 kHz through the large step sweep frequency, and then the specific resonant frequency is determined through the small step sweep frequency. As shown in Figure 23, the frequency in drive mode is 7781.1 Hz, the driving quality factor $Q_{x}$ in drive mode is 64,842, the frequency in sense mode is 7715.4 Hz, the quality factor $Q_{y}$ in sense mode is 77,154, and the frequency split is 65.7 Hz.

Figure 24 shows the results of the gyroscope along the *Z*-axis. The performance test was carried out at room temperature. The intensity of the light emitted by the tuning laser is 1 mw. The gain of the photodetector was set to 10 dB. All the data were collected by a multimeter and sent to the host computer through a serial port. Figure 24a shows that the scale factor of the gyroscope is 2.63 mv/°/s. The light intensity and the shift of the wavelength in Figure 24b cannot be measured directly. Thus, the light intensity and the deviation of the resonant wavelength can be calculated without considering the scatter loss. According to Equation (27) and the gain of the photodetector, the relationship between the angular velocity and the shift in resonant wavelength is obtained. The calculated slope from Figure 24b is 0.161 pm/°/s.

According to the MOEMS gyroscope described above, one of the most important performance indicators of the MOEMS gyroscope is its bias instability. The bias of the MOEMS gyroscope was tested on a static platform at room temperature and the test lasted 3000 s. The test results are shown in Figure 25 in form of Allan variance. By the seven-group repeatability test, the mean bias instability of the MOEMS gyroscope is 4.0399°/h and the angular random walking (ARW) value is 0.4326${}^{\circ}/\sqrt{h}$.

## 6. Conclusions

A novel MOEMS gyroscope based on a submicron-sized WGM resonator was developed. The quality factor and coupling relationship were greatly improved from adjustment of the radius of the cavity, waveguide width, and the gap between the waveguide and the cavity. Silicon nitride was finally selected as the optical section material. The theoretical value of the Q of the silicon and silicon nitride WGM resonator are 5303 and 3602. Furthermore, we combined different kinds of MEMS fabrication process and test-relevant parameters for fabrication. The sample we fabricated has a sensitivity of 2.63 mv/°/s and bias instability of 4.0399°/h, which proved that the MOEMS gyroscope can achieve a certain degree of accuracy. However, in our work, the MOEMS gyroscope with an integrated optical detection function still needs to be explored deeply, and the frequency split should be tuned by an excellent control strategy. Hence, future work will focus on further optimization and improvement for fabrication and the highly integrated optical MEMS design of the whole device.

## Figures and Tables

**Figure 1 sensors-20-07264-f001:**
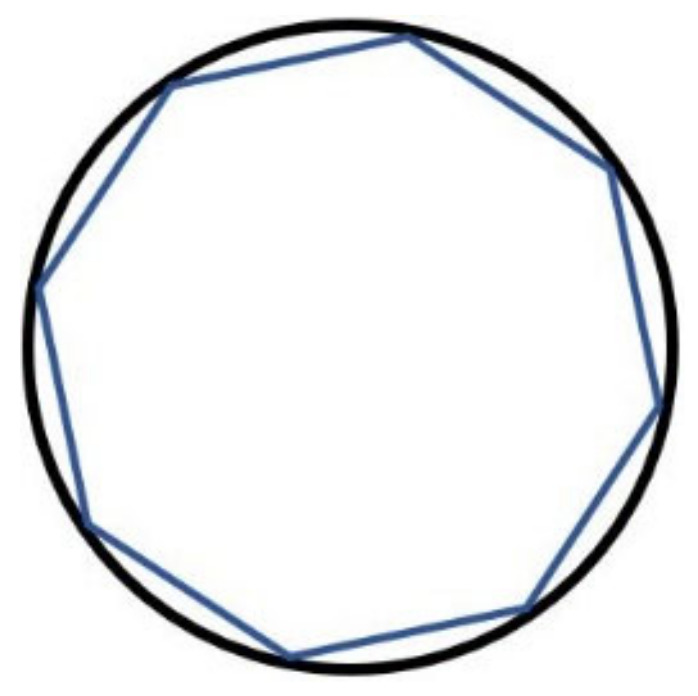
Two-dimensional planer optical path diagram of the WGM resonator.

**Figure 2 sensors-20-07264-f002:**
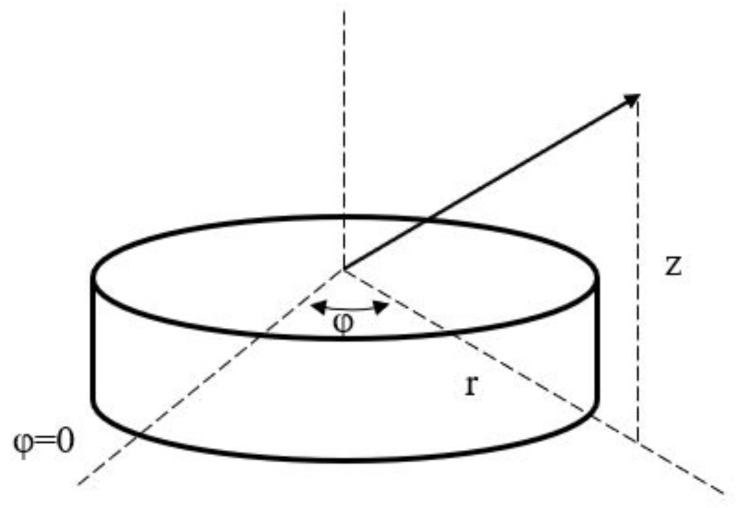
WGM resonator in a three-dimensional cylindrical coordinate system.

**Figure 3 sensors-20-07264-f003:**
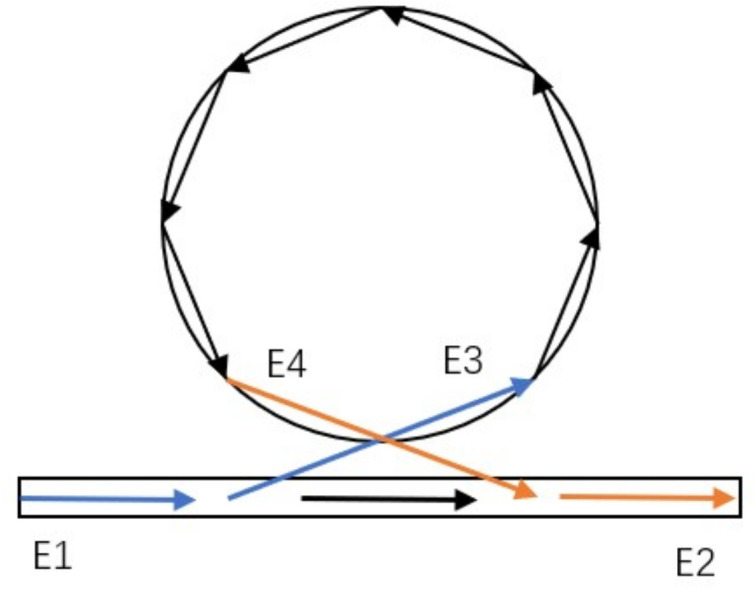
Model of cavity-waveguide coupling.

**Figure 4 sensors-20-07264-f004:**
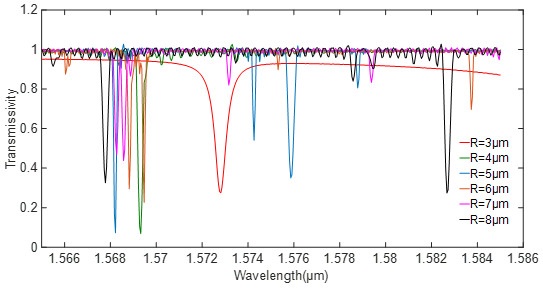
Relationship between the transmissivity and the radius of the silicon resonator.

**Figure 5 sensors-20-07264-f005:**
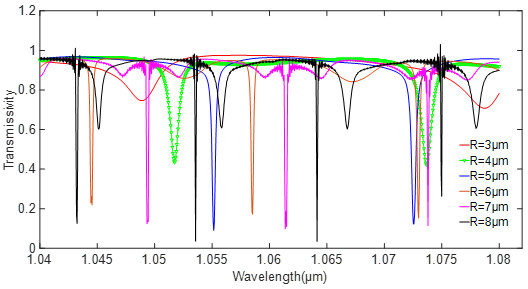
Relationship between the transmissivity and the radius of the silicon nitride resonator.

**Figure 6 sensors-20-07264-f006:**
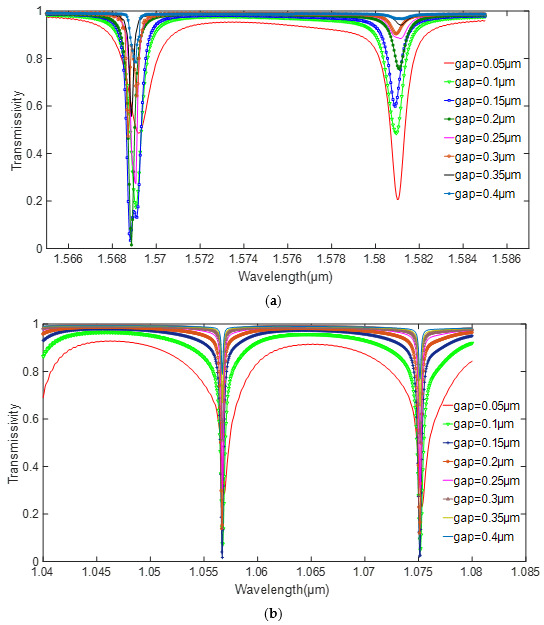
Relationship between the transmissivity and coupling gap. (**a**) silicon. (**b**) silicon nitride.

**Figure 7 sensors-20-07264-f007:**
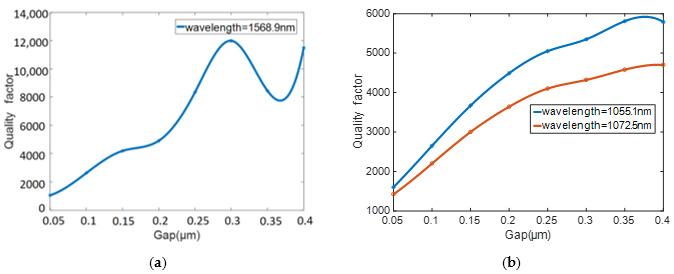
Relationship between the *Q* value and coupling gap. (**a**) silicon. (**b**) silicon nitride.

**Figure 8 sensors-20-07264-f008:**
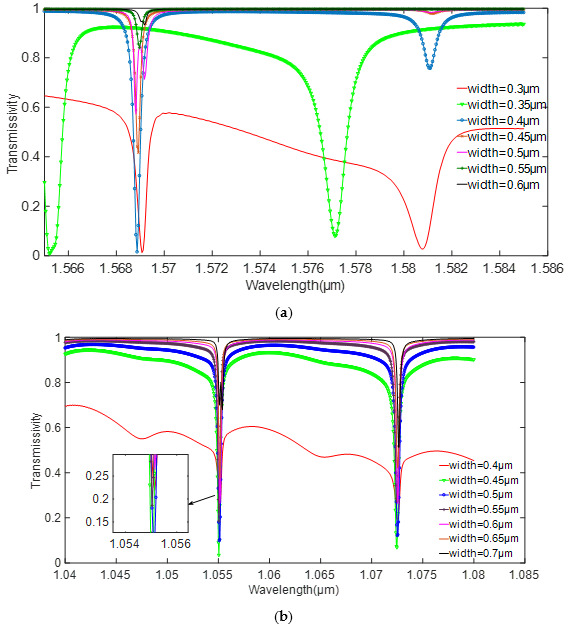
Relationship between the transmissivity and waveguide width. (**a**) silicon. (**b**) silicon nitride.

**Figure 9 sensors-20-07264-f009:**
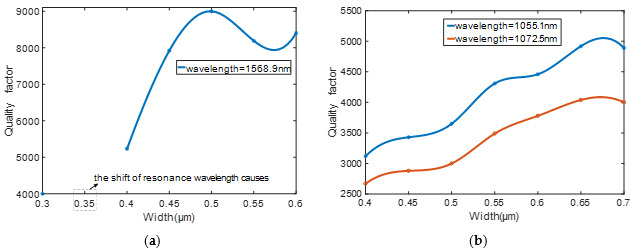
Relationship between the Q value and waveguide width. (**a**) silicon. (**b**) silicon nitride.

**Figure 10 sensors-20-07264-f010:**
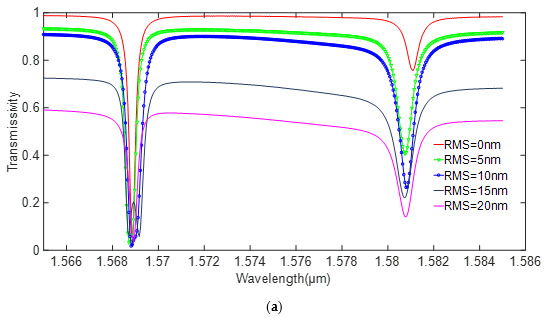
Effect of roughness on scattering loss. (**a**) silicon. (**b**) silicon nitride.

**Figure 11 sensors-20-07264-f011:**
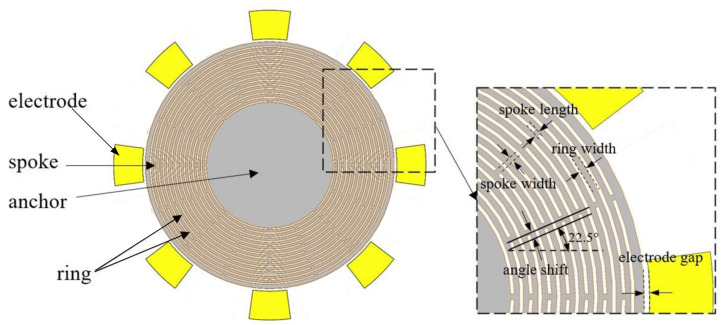
Structure of the disk resonator gyroscope.

**Figure 12 sensors-20-07264-f012:**
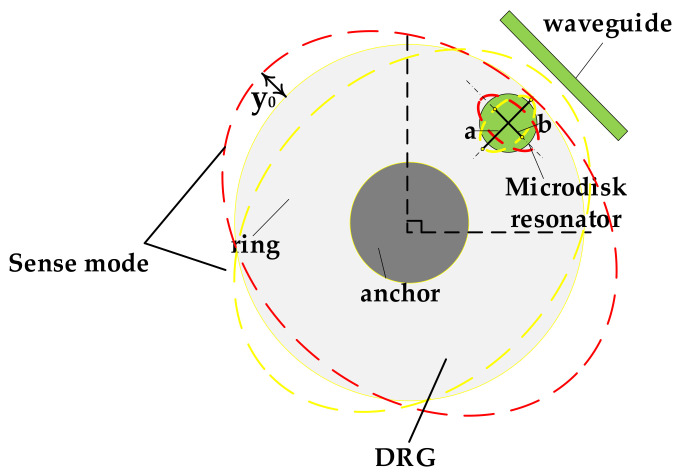
The diagram of the WGM resonator deformation.

**Figure 13 sensors-20-07264-f013:**
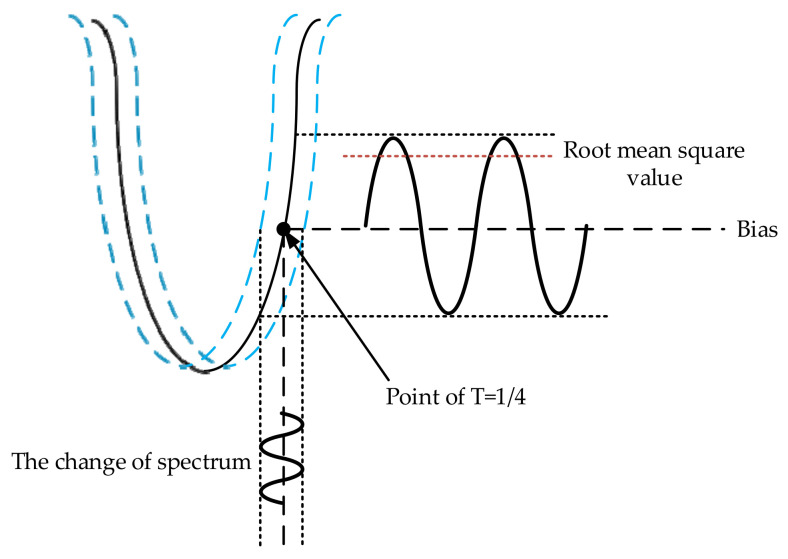
The change in the transmission spectrum.

**Figure 14 sensors-20-07264-f014:**
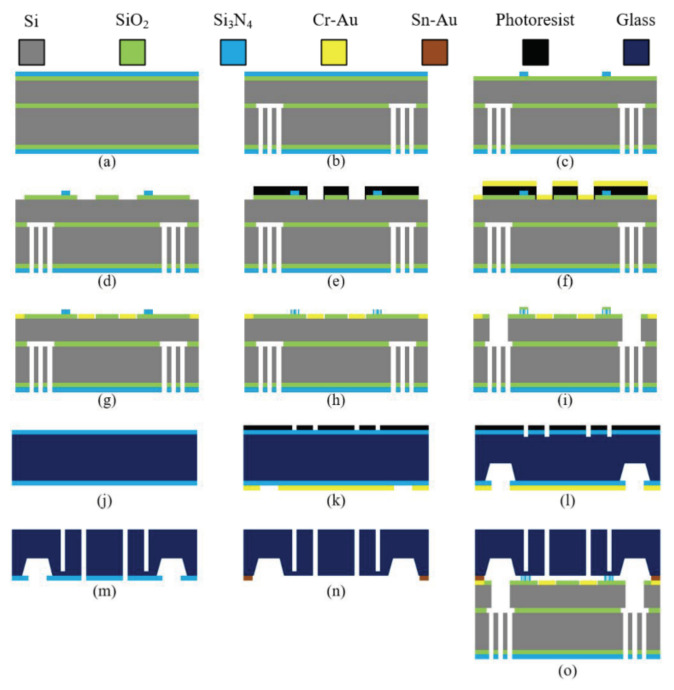
MEMS fabrication process of the gyroscope.

**Figure 15 sensors-20-07264-f015:**
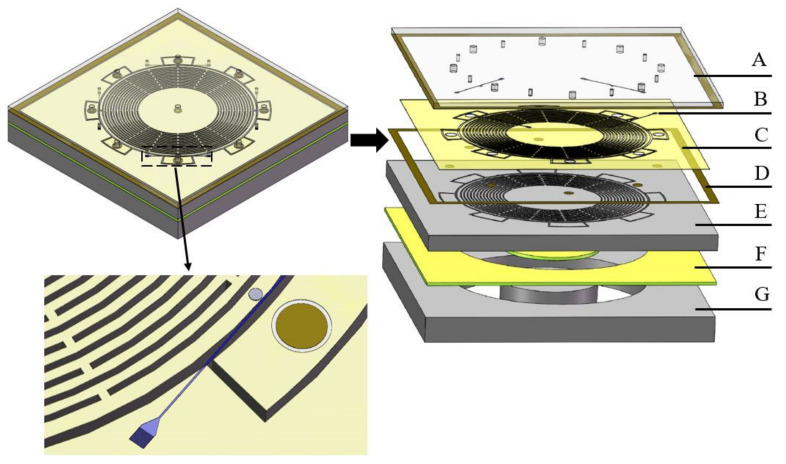
3-D structure of the MOEMS gyroscope.

**Figure 16 sensors-20-07264-f016:**
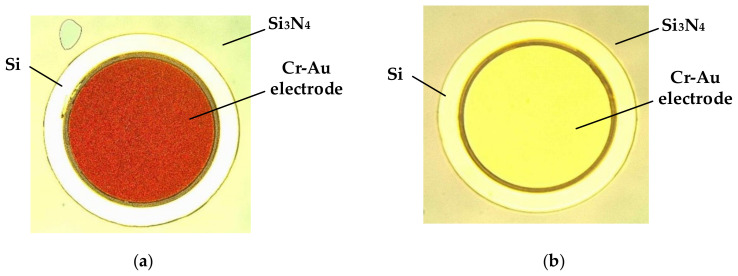
Magnetron sputtering results of a wafer after (**a**) reactive ion etching (RIE) and (**b**) buffered oxide etch (BOE).

**Figure 17 sensors-20-07264-f017:**
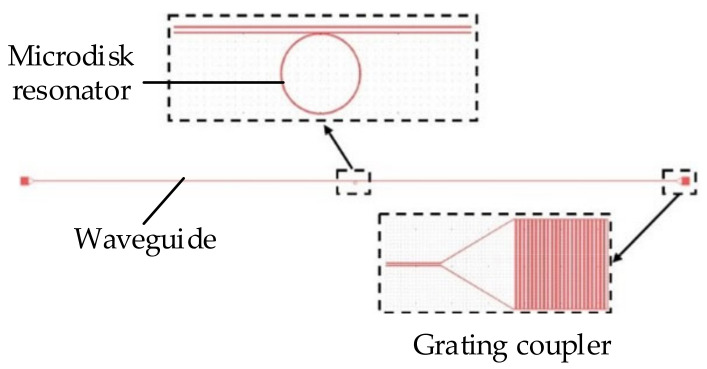
Exposure pattern of the micro-optical device.

**Figure 18 sensors-20-07264-f018:**
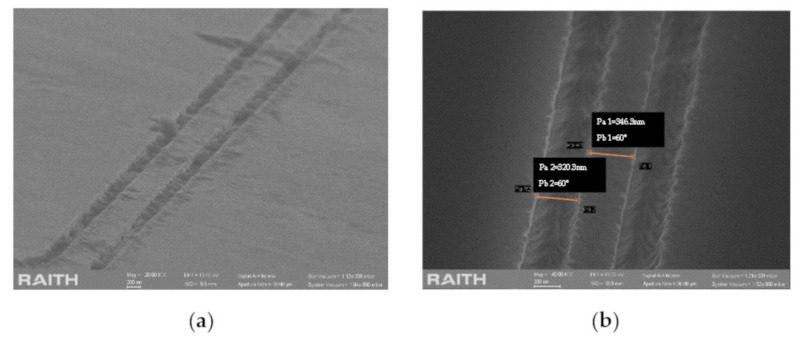
SEM image of etching of (**a**) RIE and (**b**) BOE.

**Figure 19 sensors-20-07264-f019:**
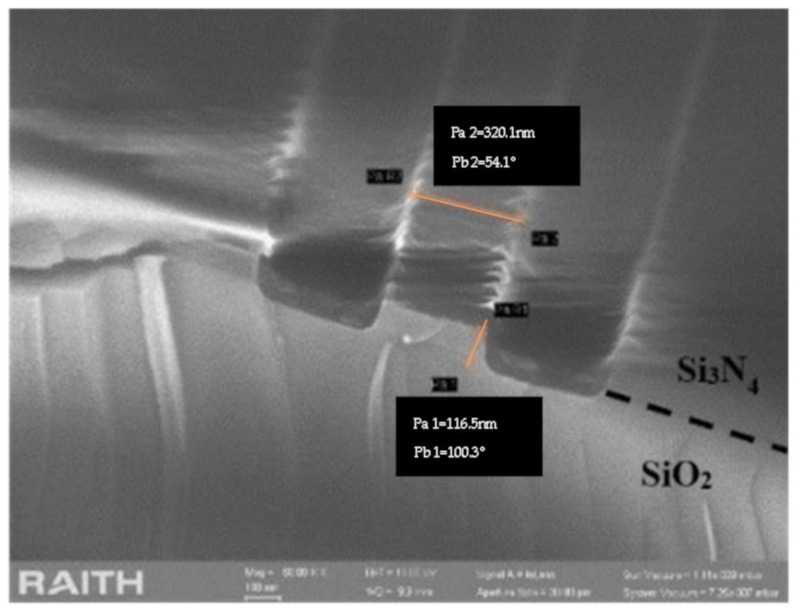
Etching result after RIE with the protection of PMMA-A4.

**Figure 20 sensors-20-07264-f020:**
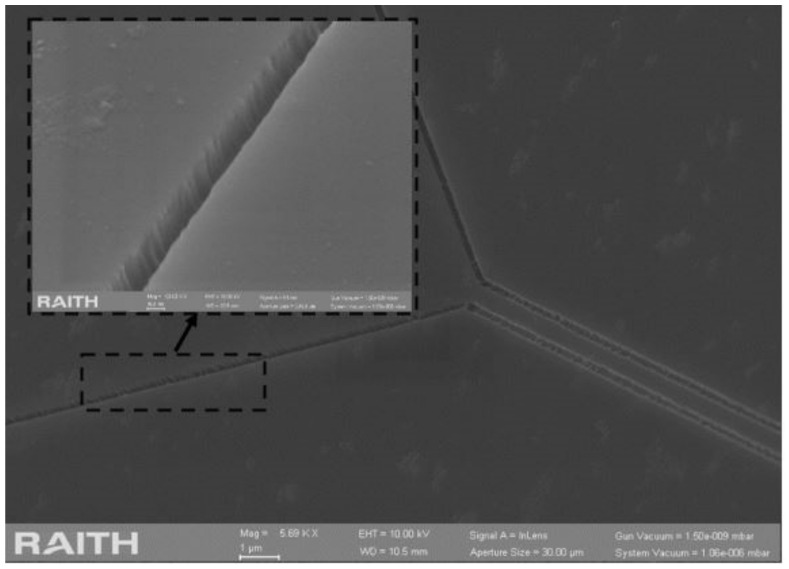
Etching result after RIE using the negative resist scheme.

**Figure 21 sensors-20-07264-f021:**
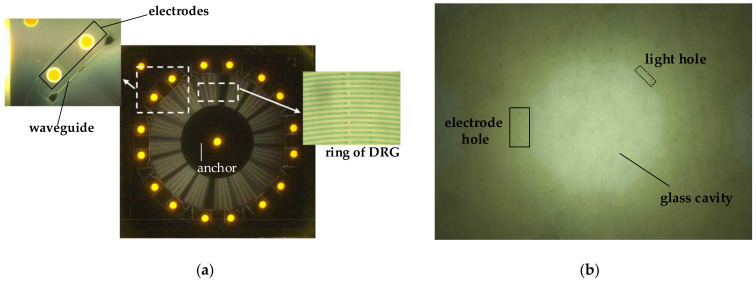
Structure of the MOEMS gyroscope. (**a**) SOI. (**b**) glass.

**Figure 22 sensors-20-07264-f022:**
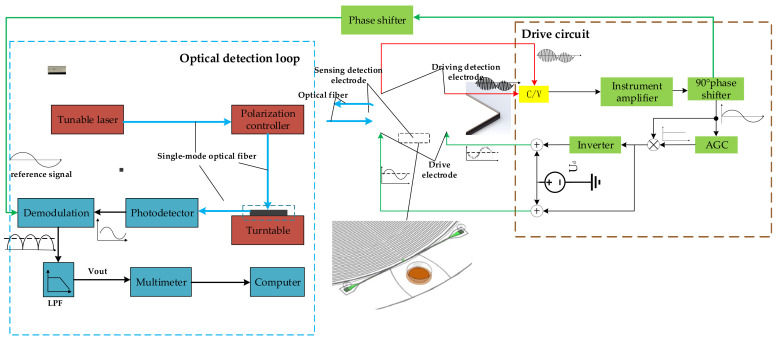
Experimental setup.

**Figure 23 sensors-20-07264-f023:**
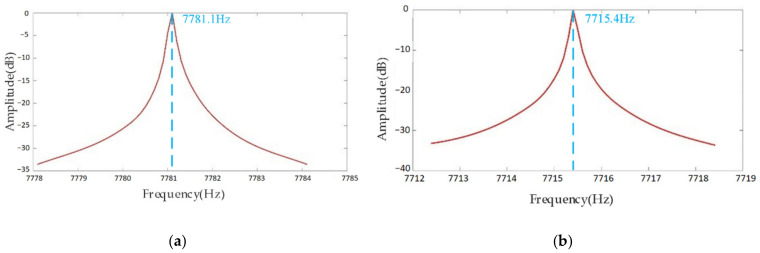
Resonance frequencies in (**a**) drive mode and (**b**) sense mode.

**Figure 24 sensors-20-07264-f024:**
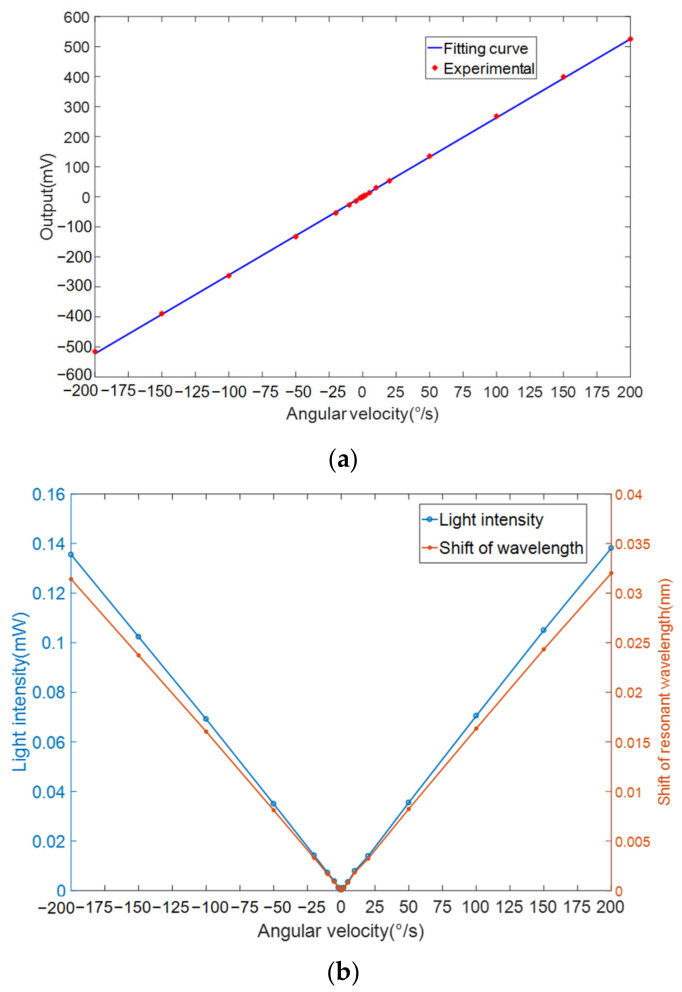
The relationship between output and angular velocity. (**a**) Voltage. (**b**) Light intensity and the shift of resonant wavelength.

**Figure 25 sensors-20-07264-f025:**
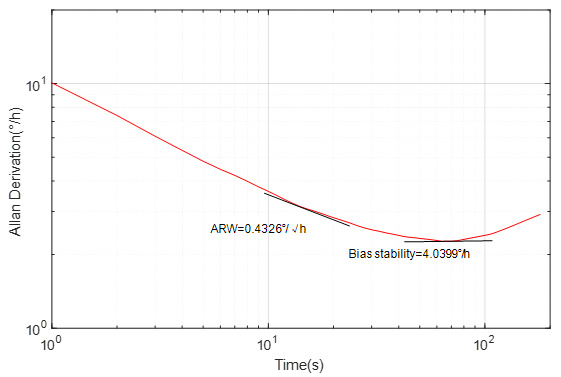
Allan derivation result.

**Table 1 sensors-20-07264-t001:** Model parameters in the simulation.

Material	Thickness (μm)	Waveguide Width (μm)	Coupling Gap (μm)	Resonator Radius (μm)
Silicon	0.2	0.3–0.6	0.05–0.4	3–8
Silicon nitride	0.3	0.4–0.7	0.05–0.4	3–8

**Table 2 sensors-20-07264-t002:** The WGM resonator parameters.

Material	Thickness (μm)	Resonator Radius (μm)	Waveguide Width (μm)	Coupling Gap (μm)	Resonant Wavelength (nm)	Q
Silicon	0.2	4	0.4	0.2	1568.9	5303
Silicon nitride	0.3	5	0.5	0.15	1055.1	3602

**Table 3 sensors-20-07264-t003:** Structure parameters of the DRG.

Structure Parameters	Value
Spoke number	16
Angle shift (Offset angle)	0.3°
Spoke width	10 μm
Spoke length	20 μm
Ring number	60 μm
Ring width	20 μm
Anchor radius	3.6 mm
Anchor height	20 μm
Electrode gap	15 μm

**Table 4 sensors-20-07264-t004:** Ion beam etching (IBE) results of different factors.

Dose Factor	Grating Width (nm)	Waveguide Width (nm)	Coupling Gap (nm)	Cavity Radius (nm)
0.4	372.1	-	-	-
0.5	358.4	612.2 *	267.8 *	-
0.6	346.8	549.4 *	212.3 *	-
0.7	312.3	498.7	149.8	4946
0.8	247.3	491.3	152.3	4935
0.9	141.7	468.9	179.6	4935
1.0	87.2 *	446.6	223.3	4924
1.1	46.7 *	424.3	224.3	4912
Desired value	360	500	150	5000

*: Not well etched. -: Not etched due to inappropriate dose.

**Table 5 sensors-20-07264-t005:** Wet etching with the protection of 8-μm-thick AZ4620 photoresist (Scheme I).

Solution	Etching Time (min)	Initial Diameter (μm)	Depth after Etching (μm)	Diameter after Etching (μm)	Aspect Ratio	Etching Rate (A/s)	Photoresist
HF	1	282	3.65	464.1	49.9:1	607.7	Broken
	2		6.78	596.6	46.4:1	565.3	Broken
	3		7.64	702.2	55:1	424.3	Broken
BOE	10		0.235	290	34:1	3.92	Intact
	15		0.423	302.7	49.1:1	4.69	Intact
	90		1.842	389.5	58.4:1	3.41	Broken

**Table 6 sensors-20-07264-t006:** Wet etching with the protection of 12-μm-thick AZ4620 photoresist (Scheme II).

Solution	Etching Time (min)	Initial Diameter (μm)	Depth after Etching (μm)	Diameter after Etching (μm)	Aspect Ratio	Etching Rate (A/s)	Photoresist
HF	3	500	7.54	591.2	12.1:1	418.9	Intact
	5		12.66	650.4	11.9:1	422	Intact
	13		34.87	936.4	12.5:1	447.1	Broken
BOE	30		0.851	518.9	22.2:1	4.72	Intact
	60		1.647	550.9	30.9:1	4.58	Intact
	90		2.373	594.9	40:1	4.39	Intact

**Table 7 sensors-20-07264-t007:** Results of wet etching with the protection of metal.

Solution	Etching Time (min)	Initial Diameter (μm)	Depth after Etching (μm)	Diameter after Etching (μm)	Aspect Ratio	Etching Rate (A/s)	Cr-Au
HF	3	6244	10.21	6298.4	5.33:1	567.2	Intact
	5		17.14	6323.3	4.63:1	571.3	Intact
	8		28.9	6339.7	3.31:1	602.1	Intact
BOE	10		35.94	6364.6	3.36:1	599	Intact
	12		40.54	6375.6	3.24:1	563.1	Intact
	15		54.67	6402.4	2.89:1	607.4	Intact

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
