# Peer review of "Optimization and Fabrication of an MOEMS Gyroscope Based on a WGM Resonator"

_sensors, 2020, doi:10.3390/s20247264_

Round 1
Reviewer 1 Report
Hi
the manuscript needs lot of editing to improve the language, figures, equations.
please see the attached comments.

Author Response
Thank you for your valuable comments. Please see the attachment.

Reviewer 2 Report
The concept is interesting and worth exploring. But the novelty of the work is unclear and the presentation is very poor.
1) Given that WGM gyroscopes are not new, the authors should distinguish their work from prior reports. What is the novelty of this WGM gyroscope?
2) The authors have also published in the design aspects of a WGM gyroscope previously. But this work is not cited in the present submission. How does this submission relate to their previous work?
3) There are many writing errors both in sentence structure, formatting (some symbols do not appear properly), spelling mistakes.
4) Symbols are often not defined (e.g. all the various parameters in relation to Q).
5) The graphs are all directly copied from Excel with no formatting, and of very poor quality. Rather than use the symbol for Lambda, the authors simply type "lambda" in the axis title. Why not simply use write "wavelength"? In one part, the authors refer to a comparison of sharpness of peaks in relation to size. But it is not possible to compare based on the curves presented with limited data points around the various peaks. In a number of the captions, the authors simply write "cont." The authors need to be better organise their graphs and use informative captions.
6) The fabricated device looks exactly like a capacitive silicon DRG, much like the ones previously reported by Tom Kenny's group at Stanford and David Horsley's group at UC Davis. This was really confusing because that is not something one would expect of a WGM optical gyroscope. All the characterisation results also appear to be using capacitive drive and sense. I can see nothing optical, not on the tests nor from the fabricated structure.
7) On the Allan Deviation, what is the output parameter that is being measured? By using the term "bias stability", once again that applies an operation of a capacitive DRG rather than a optical WGM gyroscope.
8) Assuming that the readout is optical, there are no details on sensitivity measurements to calibration the relation between the optical output and yaw rate. Information on the biasing configuration applied to disk via the electrodes is missing. What are these seven tests? Are the conditions kept the same in each instance and the authors are testing for reproducibility or is a drive parameter being tuned (please provide details).
Author Response

(The authors gave the same response as above.)

Reviewer 3 Report
This manuscript developed a MOEMS gyroscope based on a whispering gallery mode (WGM) resonator. The behavior of the mechanical properties of the resonator was studied. However, this manuscript must be improved considering the following comments:
1.-Authors must check the English style and grammar of all the manuscript.
2.-Abstract section must include the main results of the proposed prototype.
3.-Introduction section. This section must add the main scientific contribution or advantages of the proposed MOEMS gyroscope with respect to other gyroscopes.
4.-Th equation (21) must include reference.
5.-The phrase on line 174 should include more information about the optimization of the quality factors.
6.-The Figures 4-10 and 21 and 22 have poor quality. For instance, curves of Figure 4-6 and 7-8 are superimposed. The scale of these Figures must be adjusted. Figures 7 and 9 have curves with poor quality. In addition, these Figures have labels and axes thinner. The Figures should include the names of the symbols. For instance, T is transmissivity. The quality of the symbols, curves, axes, scale, and labels of all the Figures must be improved. Authors should use scientific software as Matlab to obtain the graphics.
7.-Authors should improve the discussions about the results of the Figures 6-8 and 10.
8.-Authors should add more detail information about the optimization of the MOEMS gyroscope.
9.-Section of performance test is short. This section must include more information and data of the performance test and discussion of the main results. Also, this section must include the disadvantages or challenges of the proposed device.
10.-Conslusion section must be improved considering the modifications of the revised manuscript.
11.-Authors should consider more recent references between 2018 and 2020.
Author Response

(The authors gave the same response as above.)

Round 2
Reviewer 2 Report
With the recent corrections made, the manuscript is a lot more readable and the concepts are a lot clearer.
While it is evident that the writing has been improved, there are still awkward expressions scattered throughout the manuscript starting even from the first line of the abstract. Please do another thorough check.
In both text and figure legends, the authors fail to consistently use the Greek symbol "miu" for the micro prefix and instead use "um". Please do a thorough check and make the corrections.
Line 245/246: "sense" modes (instead of "sensitive" modes")
Author Response
Thanks for your careful suggestion! In response to the deficiencies you pointed out,we did a through check for this manuscript again, including correcting some inappropriate expressions, and modifing the "μm" in the figure legends and text.
Reviewer 3 Report
This manuscript has been improved considering all the reviewer's comments.
Author Response
Thanks for your careful suggestion! We did a through check for this manuscript again, including correcting some inappropriate expressions, and modifing the "μm" in the figure legends and text. Meanwhile, we have also modified some sentence structures to make the expression clearer.